# Association between dyslipidemia and the risk of incident chronic kidney disease affected by genetic susceptibility: Polygenic risk score analysis

**Boram Weon**[1⊛], **Yunjeong Jang**[2⊛], **Jinyeon Jo**[3], **Wencheng Jin**[1,4], **Seounguk Ha**[5], **Ara Ko**[6], **Yun Kyu Oh**[1,4], **Chun Soo Lim**[1,4], **Jung Pyo Lee**[1,4], **Sungho Won**[2,3]*, **Jeonghwan Lee**[1,4]*

1 Department of Internal Medicine, Seoul National University Boramae Medical Center, Seoul, Republic of Korea, 2 Rexsoft Corporation, Seoul, Republic of Korea, 3 Department of Public Health Sciences, Institute of Health & Environment, School of Public Health, Seoul National University, Seoul, Republic of Korea, 4 Department of Internal Medicine, Seoul National University College of Medicine, Seoul, Republic of Korea, 5 Korea Medical Institute, Seoul, Republic of Korea, 6 Department of Internal Medicine, Seoul National University Hospital, Seoul, Republic of Korea

⊛ These authors contributed equally to this work.
* woogaelee@naver.com (JL); won1@snu.ac.kr (SW)

**Data Availability Statement:** The data underlying this article were accessed from the CKDGen consortium (URL: https://ckdgen.imbi.uni-freiburg.de/) or in the cited references. The UK Biobank data

## Abstract

### Background

The effect of dyslipidemia on kidney disease outcomes has been inconclusive, and it requires further clarification. Therefore, we aimed to investigate the effects of genetic factors on the association between dyslipidemia and the risk of chronic kidney disease (CKD) using polygenic risk score (PRS).

### Methods

We analyzed data from 373,523 participants from the UK Biobank aged 40–69 years with no history of CKD. Baseline data included plasma levels of total cholesterol, low-density lipoprotein cholesterol (LDL-C), high-density lipoprotein cholesterol (HDL-C), and triglyceride, as well as genome-wide genotype data for PRS. Our primary outcome, incident CKD, was defined as a composite of estimated glomerular filtration rate < 60 ml/min/1.73 m$^2$ and CKD diagnosis according to International Classification of Disease-10 codes. The effects of the association between lipid levels and PRS on incident CKD were assessed using the Cox proportional hazards model. To investigate the effect of this association, we introduced multiplicative interaction terms into a multivariate analysis model and performed subgroup analysis stratified by PRS tertiles.

### Results

In total, 4,424 participants developed CKD. In the multivariable analysis, PRS was significantly predictive of the risk of incident CKD as both a continuous variable and a categorized

for this study will be made available in the UK Biobank consortium (https://biobank.ctsu.ox.ac.uk). Data used in this study, excluding personal information, are available upon request from the corresponding author.

**Funding:** This study was supported by a multidisciplinary research grant-in-aid from the Seoul Metropolitan Government, Seoul National University (SMG-SNU) Boramae Medical Center (No.04-2022-0035). The funders had no role in study design, data collection and analysis, decision to publish, or preparation of the manuscript. Boram Weon, Yun Kyu Oh, Chun Soo Lim, Jung Pyo Lee, and Jeonghwan Lee are employed by SMG-SNU Boramae Medical Center, which provided research funding, and receives regular salary payments.

**Competing interests:** The authors have declared that no competing interests exist.

**Abbreviations:** PRS, polygenic risk score; GWAS, genome-wide association study; CKD, chronic kidney disease; eGFR, estimated glomerular filtration rate; LDL-C, low-density lipoprotein cholesterol; HDL-C, high-density lipoprotein cholesterol.

variable. In addition, lower total cholesterol, LDL-C, HDL-C, and higher triglyceride levels were significantly associated with the risk of incident CKD. There were interactions between triglycerides and intermediate and high PRS, and the interactions were inversely associated with the risk of incident CKD.

## Conclusions

This study showed that PRS presented significant predictive power for incident CKD and individuals in the low-PRS group had a higher risk of triglyceride-related incident CKD.

## Introduction

Dyslipidemia, despite its strong association with cardiovascular disease, diabetes mellitus, and hypertension, studies have shown inconsistent results regarding its impact on the onset and prognosis of CKD. In observational studies conducted on the general population, elevated levels of total cholesterol and LDL-cholesterol (LDL-C), as well as low levels of HDL-cholesterol (HDL-C), have exhibited varying associations with kidney function decline, showing significant correlations in some studies and lacking such associations in others [1–3]. Moreover, randomized controlled trials comparing kidney-related disease outcomes after the use of lipid-lowering agents such as statins, excluding patient groups with cardiovascular disease, have consistently shown that the use of these agents have no effects on the outcomes [4, 5]. Furthermore, observational studies of CKD populations have demonstrated that cholesterol and triglyceride levels are unrelated to the onset of end-stage kidney disease (ESKD), except in the low albuminuria subgroup in one study [6–8]. Currently, lipid-lowering management is recommended for CKD populations primarily for the prevention of cardiovascular disease [9]. Considering the close relationship between CKD and cardiovascular disease, these inconsistencies make it necessary to evaluate each individual's risk of CKD from dyslipidemia for proper management.

Meanwhile, susceptibilities to dyslipidemia and chronic kidney disease (CKD) are affected by genetic factors in both monogenic and polygenic ways [10–12]. A Japanese community-based cohort study identified several common genetic loci shared by patients with dyslipidemia and CKD as risk alleles [13]. Nevertheless, the effects of genetic factors on the association between dyslipidemia and kidney disease have not yet been clarified.

Genome-wide association studies (GWASs) have enabled researchers to identify numerous genetic loci associated with polygenic disorders [14]. Several GWAS have identified risk alleles associated with the development of CKD, visualized in Manhattan plots [15, 16]. Polygenic risk score (PRS) represents the sum of risk alleles weighted by the effect size of each allele, which enables us to stratify an individual's genetic susceptibility to disease as a single estimate [17, 18]. Recent studies have revealed that, when PRS constructed from a GWAS dataset is applied to another population, it could significantly predict the risk of polygenic diseases such as diabetes, cardiovascular disease, and malignant disease [19–21]. In a previous study, PRS was constructed for eGFR using summary statistics from the GWAS of CKD Genetics (CKDGen), and findings from the UK Biobank and showed that PRS had a strong association with incident CKD, ESKD, and acute kidney injury [22]. Using PRS, we can improve individualized genetic risk prediction for specific disease outcomes and investigate the interactions between genetics and other risk factors.

In this study, we hypothesized that an individual's genotypic background could modify the impact of dyslipidemia on the development of incident CKD. Therefore, we aimed to investigate the effects of genetic factors on the association between dyslipidemia and the risk of CKD. To explore this interaction, we calculated the PRS for incident CKD using summary statistics from the CKDGen Consortium and investigated the effects of lipid levels, PRS, and their interaction on incident CKD in the UK Biobank population.

## Materials and methods

### Study population

The UK Biobank is a prospective cohort with 502,516 participants aged 40–69 years recruited between 2006 and 2010. The participants were assessed at 22 centers throughout the UK, and the baseline assessment comprised a self-completed questionnaire, physical and functional measures, and sample collection. Biochemical assays and genome-wide genotyping were performed using the blood samples collected at the time of recruitment [23]. The follow-up data were obtained through linkages to national datasets until September 5, 2019.

Individuals who withdrew from enrollment during the follow-up period (N = 57), were of non-white ethnicity (N = 29,810), were related to other participants within 3rd degree (N = 60,987) or had missing data on serum creatinine level (N = 33,148) or genotype (N = 15,233) were excluded. In addition, participants were excluded if they met the criteria for CKD at baseline, which is eGFR < 60 ml/min/1.73 $m^2$ or CKD diagnosis according to International Classification of Disease-10 (ICD-10) codes (N = 10,883), if they had already received kidney transplantation (N = 359) or were on dialysis (N = 282). The final cohort comprised 373,523 participants (Fig 1).

### Ethical considerations

The participants provided written informed consent, and participation in the UK Biobank was voluntary. The study was conducted in accordance with the principles of the Declaration of Helsinki. The investigators applied for the UK Biobank data (APPLICATION No. 41056) after obtaining the IRB approval (Seoul National University IRB No. SNU 16-03-076). The Institutional Review Boards of the participating hospitals (Seoul National University Boramae Medical Center: 07-2022-33) approved the study protocol and waived the need for additional informed consent for patient participation. The data for this study was last updated on March 8, 2023.

### Baseline assessments and measurements

Assessment at the time of recruitment included a self-completed questionnaire, an interview, physical measurements, and blood sample collection [23]. Using a questionnaire and an interview, information was obtained regarding demographic features (age and sex), lifestyle factors (alcohol intake frequency and smoking status), and self-reported medical history and medications. Physical measures included blood pressure and body mass index (BMI). Biochemical data such as serum creatinine, non-fasting lipid profiles, albumin, C-reactive protein (CRP), HbA1c, and genotype data were derived from blood samples collected at baseline. Plasma lipid profiles encompassed total cholesterol, HDL-C, LDL-C, and triglycerides. We also assessed baseline comorbidities of diabetes mellitus, hypertension, and dyslipidemia according to the ICD-10 codes, current medication for each disease, as well as other estimates such as HbA1c, blood pressure, and lipid levels.

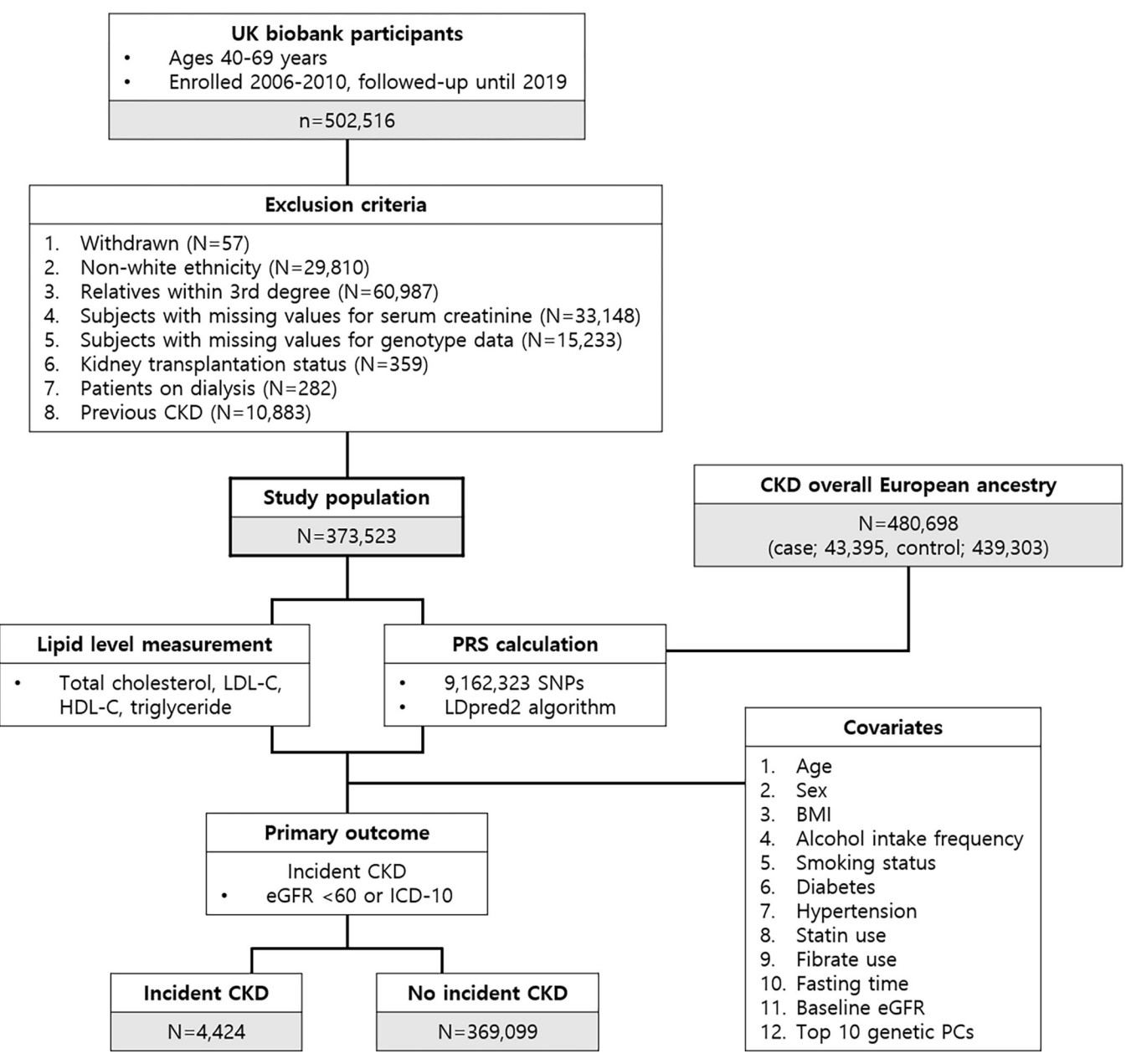

**Fig 1. Study design.** Data from the UK Biobank participants enrolled from 2006 to 2010 and followed up until 2019 were utilized. A total of 373,523 participants were included in this study. The PRS was calculated using summary statistics of GWAS from patients with CKD European ancestry. The primary outcome of incident CKD was defined as a follow-up eGFR < 60 or a new registration of CKD according to the ICD-10 code.

## Polygenic risk score construction

The base GWAS set for the construction of PRS was obtained from meta-analysis data of the CKDGen consortium [14]. We used summary statistics of CKD overall European ancestry, which included 480,697 individuals (41,395 cases and 439,303 controls). This GWAS summary data consisted of 9,162,323 SNPs with MAF ≥ 0.5%, and the UK Biobank shared 3 898 527 SNPs of the total. There was no participant overlap between the base GWAS set and the UK Biobank. For validation, 50 000 individuals were randomly selected independently of the training set. We constructed PRS using five different algorithms: clumping and thresholding (C+T)

and pruning and thresholding (P+T) using PLINK [17], LDpred2 [24], LASSOsum [25] and PRScs [26]. In the PRS tuning process, we finally chose a model using LDpred2 as an optimal PRS based on the highest correlation (R), significance (p-value), and the lowest Akaike information criterion (AIC) in regression. The comparison of each PRS algorithm is visualized in S1 Table. Genotype data was managed with PLINK and ONETOOL [27].

## Outcomes ascertainment

In the UK Biobank protocol, long-term follow-up is performed through linkages with national datasets of health records obtained with consent, including hospital records, primary care records, and death and cancer registries [23]. The outcome of this study was incident CKD during follow-up, defined as a composite of eGFR $< 60$ mL/min/1.73 $m^2$ and the development of CKD based on ICD-10 codes. We calculated eGFR using the Chronic Kidney Disease Epidemiology Collaboration (CKD-EPI) equation based on age, race, sex, and serum creatinine levels [28].

## Statistical analysis

We assessed the associations of dyslipidemia and genetic factors with incident CKD using the Cox proportional hazards model. Continuous variables are presented as mean ± standard (SD) deviation, and categorical variables are presented as numbers (percentages). We used the Kolmogorov-Smirnov test for normality of the distribution. Hazard ratios (HRs) and 95% confidence intervals (CIs) were estimated by adjusting for age, sex, BMI, alcohol intake frequency, smoking status, comorbidities (history of diabetes and hypertension), statin use, fibrate use, fasting time, baseline eGFR, and the top 10 genetic principal components (PCs). We evaluated the effects of PRS both as a continuous and categorized variable in tertiles. Lipid profiles of total cholesterol, HDL-C, LDL-C, and triglycerides were standardized by subtracting the mean and dividing by the standard deviation and presented as HR per 1-SD increase in lipid level as a continuous variable. Lipid profiles were also categorized as binary variables according to the optimal levels recommended in the 2015 Korean Guidelines for the Management of Dyslipidemia [29].

To evaluate the interactions between dyslipidemia and genetic factors, we introduced a multiplicative interaction term between lipid levels and PRS into a Cox regression model. Finally, we performed a subgroup analysis of the association between lipid levels and incident CKD stratified by PRS tertiles to evaluate the differential effects of lipid levels according to genetic risk. All analyses were performed using the R software, version 3.6.3 (R Foundation) and Rex [30]. All P values were two-sided, and $P < 0.05$ was considered statistically significant.

# Results

## Baseline characteristics

Our study population included 373,523 participants, of whom 4,424 developed incident CKD during a median follow-up of 10.7 years. Table 1 shows the baseline characteristics of the study participants according to the development of incident CKD. The mean age of the participants was 56.57 ± 8.00 years, and 54.01% were female. The mean baseline creatinine level and eGFR were 0.80 ± 0.15 mg/dL and 91.66 ± 11.88 mL/min/1.73 $m^2$ respectively. Of the study population, 16.83% had dyslipidemia and 14.75% were using statins. Distribution of lipid levels is shown in S1 Fig.

Individuals with incident CKD during follow-up tended to be older, male, ever-smokers, with a higher BMI and have a higher level of serum creatinine level at baseline. Participants

**Table 1. Baseline characteristics of study population according to the incident CKD during follow-up period.**

| Variable | Total | No incident CKD | Incident CKD | P-value | Missing data |
|---|---|---|---|---|---|
| | (N = 373,523) | (N = 369,099) | (N = 4,424) | | N (%) |
| Age (years) | 56.57±8.00 | 56.51±8.00 | 62.17±5.98 | <**0.001** | 0 (0%) |
| Sex, n(%) | | | | <**0.001** | 0 (0%) |
| Female | 201731 (54.01%) | 199619 (54.08%) | 2112 (47.74%) | | |
| Male | 171792 (45.99%) | 169480 (45.92%) | 2312 (52.26%) | | |
| Body mass index | 27.34±4.74 | 27.31±4.72 | 29.74±5.61 | <**0.001** | 1176 (0.31%) |
| Alcohol intake frequency, n(%) | | | | <**0.001** | 286 (0.08%) |
| Daily or almost daily | 80075 (21.45%) | 79259 (21.49%) | 816 (18.47%) | | |
| Three or four times a week | 89826 (24.07%) | 89042 (24.14%) | 784 (17.75%) | | |
| Once or twice a week | 97769 (26.19%) | 96704 (26.22%) | 1065 (24.11%) | | |
| One to three times a month | 41413 (11.10%) | 40912 (11.09%) | 501 (11.34%) | | |
| Special occasions only | 39570 (10.60%) | 38844 (10.53%) | 726 (16.44%) | | |
| Never | 24584 (6.59%) | 24059 (6.52%) | 525 (11.89%) | | |
| Smoking status, n(%) | | | | <**0.001** | 1287 (0.34%) |
| Never | 201352 (54.09%) | 199423 (54.21%) | 1929 (43.86%) | | |
| Previous | 131742 (35.39%) | 129801 (35.29%) | 1941 (44.13%) | | |
| Current | 39142 (10.52%) | 38614 (10.50%) | 528 (12.01%) | | |
| Baseline comorbidities, n(%)† | | | | | |
| Diabetes | 17200 (4.60%) | 16315 (4.42%) | 885 (20.00%) | <**0.001** | 0(0%) |
| Hypertension | 86187 (23.07%) | 84319 (22.84%) | 1868 (42.22%) | <**0.001** | 0(0%) |
| Dyslipidemia | 62871 (16.83%) | 60892 (16.50%) | 1979 (44.73%) | <**0.001** | 0(0%) |
| Medication, n(%)† | | | | | |
| Diabetes | 9818 (2.63%) | 9222 (2.50%) | 596 (13.47%) | <**0.001** | 0(0%) |
| Hypertension | 56424 (15.11%) | 54459 (14.75%) | 1965 (44.42%) | <**0.001** | 0(0%) |
| Statin use, n(%) | 55088 (14.75%) | 53352 (14.45%) | 1736 (39.24%) | <**0.001** | 0 (0%) |
| Fibrate use, n(%) | 731 (0.20%) | 683 (0.19%) | 48 (1.08%) | <**0.001** | 0 (0%) |
| Serum creatinine (mg/dL) | 0.80±0.15 | 0.80±0.15 | 0.93±0.16 | <**0.001** | 0 (0%) |
| eGFR (ml/min/1.73m2) | 91.66±11.88 | 91.83±11.77 | 77.74±12.30 | <**0.001** | 0 (0%) |
| Lipid levels (mg/dL) | | | | | |
| Total cholesterol | 221.24±43.99 | 221.45±43.88 | 204.20±49.14 | <**0.001** | 103 (0.03%) |
| LDL-C | 138.24±33.48 | 138.38±33.41 | 126.40±36.44 | <**0.001** | 715 (0.19%) |
| HDL-C | 56.32±14.81 | 56.38±14.80 | 51.12±14.40 | <**0.001** | 31643 (8.47%) |
| Triglyceride | 154.43±90.33 | 154.14± 90.13 | 179.16±102.94 | <**0.001** | 163 (0.04%) |
| Fasting time (hours) | 3.76±2.39 | 3.76±2.38 | 3.95±2.56 | <**0.001** | 80 (0.02%) |
| Polygenic risk score tertile | | | | <**0.001** | 0 (0%) |
| Low | 124508 (33.33%) | 123370 (33.42%) | 1138 (25.72%) | | |
| Intermediate | 124507 (33.33%) | 123066 (33.34%) | 1441 (32.57%) | | |
| High | 124508 (33.33%) | 122663 (33.23%) | 1845 (41.70%) | | |

CKD, chronic kidney disease; eGFR, estimated glomerular filtration rate; LDL-C, low density lipoprotein cholesterol; HDL-C, high density lipoprotein cholesterol

Data was reported as mean ± SD for continuous variables and n(%) for categorical variables.

P-value was computed by t-test for continuous variables and chi-square test or Fisher's exact test for categorical variables as appropriate.

†Number of samples with each comorbidity or taking medication is given.

who developed incident CKD had a higher proportion of those with baseline dyslipidemia and who use lipid-lowering agents. The proportion of participants in the high PRS tertile was higher among those with incident CKD.

## Impacts of each lipid levels and polygenic risk score on incident chronic kidney disease

Associations between PRS and incident CKD and between lipid levels and incident CKD using a multivariable Cox regression model including PRS or each lipid level and other covariates are shown in Table 2. PRS was significantly predictive of the risk of incident CKD as both a continuous variable (HR, 1.076; 95% CI, 1.043–1.110) and tertile. Results of the analysis using unadjusted models are shown in S2 Table.

In analyses with models including each lipid level separately, a 1-SD increase in triglycerides (HR, 1.082; 95% CI, 1.052–1.113) and a decrease in HDL-C (HR, 0.876; 95% CI, 0.840–0.913) significantly increased the risk of incident CKD. Interestingly, total cholesterol (HR, 0.896; 95% CI, 0.865–0.929) and LDL-C (HR 0.896; 95% CI, 0.865–0.929) were inversely associated with the risk of CKD. Restricted cubic spline curves for lipid levels are shown in S2 Fig. In the subgroup analysis stratified by statin use (S3 Table), these inverse relationships were more prominent in the group without statin use. We further analyzed the baseline characteristics of participants without statin use or a history of dyslipidemia, stratified by LDL-C and total cholesterol levels (S4 Table). Participants with lower LDL-C and total cholesterol levels had higher proportion of diabetes, higher CRP levels, and lower serum albumin levels at baseline.

## Interaction between lipid levels and polygenic risk score for incident chronic kidney disease

The interactions between lipids and PRS were examined using a multivariable analysis model with each lipid level, PRS tertile (the lowest tertile was a reference category), and a multiplicative interaction term between them (Table 3). In this analysis, triglyceride level and

**Table 2. Multivariable analysis using Cox regression model for incident CKD.**

| Variables | HR (95% CI) | | P-value |
|---|---|---|---|
| **Polygenic risk score [a]** | | | |
| PRS (continuous variable) | 1.076 | (1.043–1.110) | **<0.001** |
| PRS tertile | | | |
|   Low | | Ref. | |
|   Intermediate | 1.091 | (1.009–1.181) | **0.030** |
|   High | 1.204 | (1.116–1.299) | **<0.001** |
| **Lipid levels [b]** | | | |
| Total cholesterol | 0.896 | (0.865–0.929) | **<0.001** |
| LDL-C | 0.896 | (0.865–0.929) | **<0.001** |
| HDL-C | 0.876 | (0.840–0.913) | **<0.001** |
| Triglyceride | 1.082 | (1.052–1.113) | **<0.001** |

LDL-C, low density lipoprotein cholesterol; HDL-C, high density lipoprotein cholesterol; PRS, polygenic risk score; HR, hazard ratio; CI, confidence interval; SE, standard error; Ref., reference category.

Hazard ratios were reported as per 1-SD change for lipid levels.

a: Models including polygenic risk score (continuous variable or tertile). Adjusted for age, sex, BMI, alcohol intake frequency, smoking status, comorbidities (diabetes, hypertension), statin use, fibrate use, fasting time, baseline eGFR, and top 10 genetic principal components.

b: Models including each lipid level (separate model for each lipid category). Adjusted for age, sex, BMI, alcohol intake frequency, smoking status, comorbidities (diabetes, hypertension), statin use, fibrate use, fasting time and baseline eGFR

**Table 3. Multivariable analysis for interaction between lipid levels and PRS.**

| Variables | HR (95% CI) | | P-value |
|---|---|---|---|
| Total cholesterol | 0.877 | (0.827–0.931) | <**0.001** |
| Intermediate PRS | 1.098 | (1.009–1.194) | **0.029** |
| High PRS | 1.222 | (1.127–1.325) | <**0.001** |
| Intermediate PRS * Total cholesterol | 1.015 | (0.944–1.092) | 0.684 |
| High PRS * Total cholesterol | 1.039 | (0.969–1.114) | 0.280 |
| LDL-C | 0.876 | (0.825–0.931) | <**0.001** |
| Intermediate PRS | 1.095 | (1.007–1.190) | **0.034** |
| High PRS | 1.224 | (1.129–1.326) | <**0.001** |
| Intermediate PRS * LDL-C | 1.008 | (0.935–1.086) | 0.840 |
| High PRS * LDL-C | 1.047 | (0.975–1.124) | 0.205 |
| HDL-C | 0.820 | (0.762–0.881) | <**0.001** |
| Intermediate PRS | 1.123 | (1.028–1.226) | **0.010** |
| High PRS | 1.219 | (1.119–1.328) | <**0.001** |
| Intermediate PRS * HDL-C | 1.123 | (1.026–1.228) | **0.012** |
| High PRS * HDL-C | 1.073 | (0.985–1.170) | 0.107 |
| Triglyceride | 1.152 | (1.094–1.212) | <**0.001** |
| Intermediate PRS | 1.121 | (1.033–1.215) | **0.006** |
| High PRS | 1.226 | (1.133–1.327) | <**0.001** |
| Intermediate PRS * Triglyceride | 0.899 | (0.838–0.965) | **0.003** |
| High PRS * Triglyceride | 0.934 | (0.874–0.997) | **0.041** |

LDL-C, low density lipoprotein cholesterol; HDL-C, high density lipoprotein cholesterol; PRS, polygenic risk score; HR, hazard ratio; CI, confidence interval; SE, standard error.

Hazard ratios were reported as per 1-SD change for lipid levels.

Low PRS group was a reference category.

Models including each lipid level, PRS tertile, and multiplicative interaction term between them. Adjusted for age, sex, BMI, alcohol intake frequency, smoking status, comorbidities (diabetes, hypertension), statin use, fibrate use, fasting time, baseline eGFR, the top 10 genetic principal components.

intermediate (HR, 0.899; 95% CI, 0.838–0.965) and high PRS (HR, 0.934; 95% CI 0.874–0.997) tertiles showed significant interactions, which weakened the effect of triglyceride level on the risk of incident CKD.

Similarly, in the subgroup analysis stratified by PRS tertiles (Table 4), the effect of 1-SD higher triglyceride level was significant only in the group with the lowest genetic risk (HR, 1.136; 95% CI, 1.010–1.277). In contrast, the effects of 1-SD lower total cholesterol, LDL-C, and HDL-C levels were significant, except in the group with the lowest genetic risk. Fig 2 shows a forest plot that visualizes the differential impact of lipids according to PRS tertiles.

## Discussion

In this study, we evaluated the significance of an individual's genetic susceptibility, represented by constructed PRS and individual lipid level, for CKD development. Higher triglyceride and lower total cholesterol, LDL-C, and HDL-C levels increased the risk of CKD, and the effect of lower total cholesterol and LDL-C levels on incident CKD was more prominent in the population without statin therapy. We also identified a significant interaction between triglyceride levels and PRS, which contributed to an increased risk of triglyceride-induced CKD in the low-PRS group.

**Table 4. Subgroup analysis using Cox regression model for incident CKD according to PRS tertiles.**

| | Low PRS (N = 124,508) | | | Intermediate PRS (N = 124,507) | | | High PRS (N = 124,508) | | |
|---|---|---|---|---|---|---|---|---|---|
| | HR (95% CI) | | P-value | HR (95% CI) | | P-value | HR (95% CI) | | P-value |
| Total cholesterol | 0.953 | (0.834–1.088) | 0.475 | 0.870 | (0.817–0.926) | **<0.001** | 0.875 | (0.790–0.970) | **0.011** |
| PRS | 1.014 | (0.895–1.149) | 0.831 | 1.039 | (0.828–1.304) | 0.739 | 1.040 | (0.955–1.133) | 0.369 |
| PRS * Total cholesterol | 1.034 | (0.928–1.151) | 0.548 | 1.004 | (0.823–1.224) | 0.971 | 1.026 | (0.951–1.107) | 0.502 |
| LDL-C | 0.957 | (0.836–1.095) | 0.518 | 0.867 | (0.814–0.923) | **<0.001** | 0.897 | (0.808–0.996) | **0.041** |
| PRS | 1.014 | (0.895–1.149) | 0.827 | 1.057 | (0.843–1.325) | 0.633 | 1.033 | (0.948–1.125) | 0.458 |
| PRS * LDL-C | 1.047 | (0.939–1.168) | 0.407 | 1.034 | (0.844–1.266) | 0.747 | 1.012 | (0.936–1.094) | 0.764 |
| HDL-C | 0.903 | (0.767–1.064) | 0.222 | 0.909 | (0.846–0.977) | **0.009** | 0.834 | (0.736–0.945) | **0.004** |
| PRS | 1.011 | (0.885–1.155) | 0.871 | 1.013 | (0.802–1.281) | 0.911 | 1.043 | (0.952–1.143) | 0.363 |
| PRS * HDL-C | 1.062 | (0.927–1.216) | 0.387 | 0.966 | (0.761–1.227) | 0.779 | 1.034 | (0.942–1.134) | 0.483 |
| Triglyceride | 1.136 | (1.010–1.277) | **0.034** | 1.039 | (0.987–1.093) | 0.141 | 1.038 | (0.944–1.141) | 0.444 |
| PRS | 0.997 | (0.884–1.125) | 0.963 | 1.051 | (0.846–1.307) | 0.652 | 1.019 | (0.936–1.109) | 0.668 |
| PRS * Triglyceride | 1.000 | (0.904–1.106) | 0.996 | 0.957 | (0.779–1.175) | 0.675 | 1.038 | (0.964–1.117) | 0.324 |

LDL-C, Low density lipoprotein cholesterol; HDL-C, High density lipoprotein cholesterol; PRS, polygenic risk score; HR, Hazard ratio; CI, confidence interval; SE, standard error.

Hazard ratios were reported as per 1-SD change for lipid levels.

Models including each lipid level, PRS as continuous variable, and multiplicative interaction term between them. Adjusted for age, sex, BMI, alcohol intake frequency, smoking status, comorbidities (diabetes, hypertension), statin use, fibrate use, fasting time, baseline eGFR, and top 10 genetic principal components

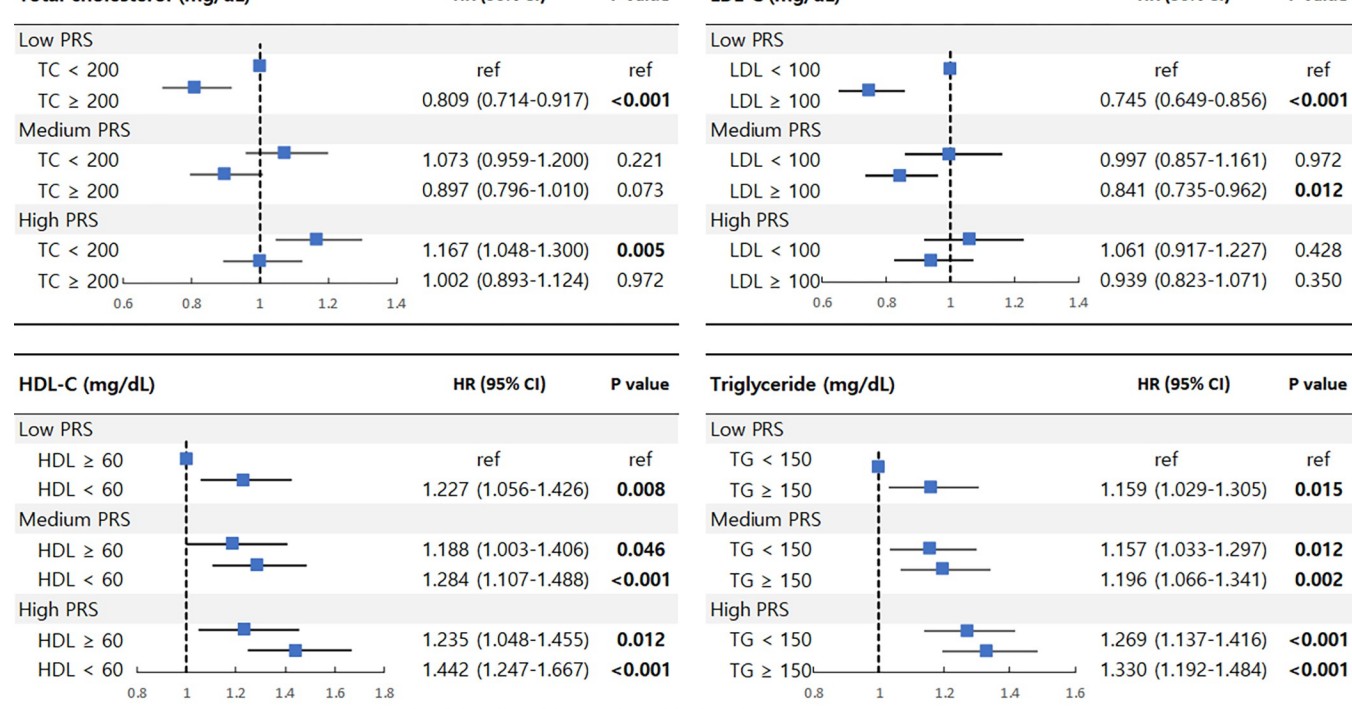

**Fig 2. Forest plot visualizing differential impacts of lipids on incident CKD according to genetic risk.** Multivariable analyses with model including each lipid as binary variable and PRS as tertiles were performed. Each model was adjusted for age, sex, BMI, alcohol intake frequency, smoking status, diabetes, hypertension, statin use, fibrate use, fasting time, baseline eGFR, the top 10 genetic principal components.

Dyslipidemia is a major risk factor for cardiovascular disease and known to be associated with the development of CKD [31]. Lipid management is recommended for patients with CKD to reduce cardiovascular morbidity and mortality [9]; however, studies are inconsistent about the effects of dyslipidemia on renal outcomes. Many general population-based studies have shown associations between lipid status and development of CKD; however, the results vary among the studies. In a study conducted among a cohort of apparently healthy men [1], high total cholesterol and low HDL-C levels were associated with increased risk of kidney dysfunction, while another community-based cohort study showed that low HDL-C level was predictive of kidney disease, but total cholesterol was not [2]. On the other hand, the role of low HDL-C level in incident CKD was rejected in a retrospective study of a Chinese population; however, high total cholesterol, LDL-C, and triglyceride levels were significantly associated with incident CKD [3].

Meanwhile, the association between dyslipidemia and renal outcomes varies depending on the characteristics of the target population, and these associations are attenuated in studies with CKD populations. In two cohort studies of patients with CKD, cholesterol and triglyceride levels were not associated with progression to ESKD [6, 8]. The other CKD population-based study also showed no independent association between lipid levels and progression of kidney disease; however, in subgroups with low levels of proteinuria, low total cholesterol and LDL-C levels were associated with an increased risk of renal function decline [7].

These inverse relationships between cholesterol and kidney disease outcomes were also present in our results and were prominent in the subgroup without baseline statin use. In this subgroup, participants with lower LDL-C and total cholesterol levels, who had a higher risk of incident CKD, tended to have higher CRP levels, lower serum albumin levels, and lower BMI. This tendency is similar to the paradoxical relationship between cholesterol levels and mortality in dialysis patients, which is an inverse association between cholesterol and mortality due to the cholesterol-lowering effect of systemic inflammation and malnutrition [32]. In addition, the proportion of participants with baseline diabetes was significantly higher among those with lower LDL-C and total cholesterol levels. It is known that LDL-C in patients with type 2 diabetes does not increase in concentration but increases in atherogenic potential through qualitative modification [33]. Therefore, we can assume that LDL-C may affect renal outcomes without increasing its levels in patients with diabetes.

Both dyslipidemia and CKD are affected by genetic risk factors in polygenic and monogenic ways [10–12], and one study revealed several shared risk genetic loci for both diseases [13]. However, to the best of our knowledge, no study has investigated the effects of genetic factors on the relationship between dyslipidemia and CKD. PRS enables us not only to evaluate an individual's risk factors for disease outcome but also to investigate the interaction between genetic and other risk factors. In a study that constructed a PRS for kidney diseases using 1.5 million SNPs from GWAS data of the UK Biobank and CKDGen meta-analysis, the PRS was sufficiently strong to capture the risk of incident kidney diseases and showed a significant association with circulating proteomes mediated by eGFR [22]. A recent study showed that the PRS for CKD had reproducible performance across different ethnicities and demonstrated additive effects between the PRS and monogenic APOL1 mutations on the risk of CKD [10]. Our study identified an interaction between triglyceride levels and genetic factors that weakens the effect of hypertriglyceridemia on incident CKD. Although the mechanisms explaining these interactions remain to be clarified, our study is the first step toward introducing an individualized therapeutic approach for dyslipidemia, considering the genetic risk of kidney disease.

Despite its various advantages, this study has several limitations. First, the study population in the UK Biobank was limited to white individuals who live in UK. Because genetic effects on

incident CKD can vary according to ethnicity, it is necessary to confirm whether the effects of PRS, lipid profiles, and their interactions on CKD are the same in other populations or ethnicities. Second, it is possible that the participants' baseline lipid levels were not accurately reflected because blood samples were not collected during fasting. When constructing the UK biobank, researchers did not ask participants to fast before assessment because, in the pilot study, there was not much discrepancy in the reported hours from the last meal between the groups who required fasting and those who did not, which lasted 4 to 5 hours in each group. In addition, because clinical and biochemical data were not obtained through regular visits during the follow-up period, the CKD incidence may have been underestimated. Furthermore, information regarding the timing of diagnosis of diabetes and hypertension was not available from the UK Biobank, variables regarding the duration of exposure to hyperglycemia and high blood pressure, which are important in the development of kidney disease, could not be applied to the analysis. Further cohort studies, including more precise and regular assessments of clinical and lifestyle data incorporating temporal information, are warranted. Finally, in the present study, since the mechanism underlying the interaction between triglycerides and genetic factors is not elucidated, additional investigations are necessary to explore the specific genetic loci or pathways that mediate the interaction.

## Conclusion

The study showed a significant association between the PRS for CKD and the risk of incident CKD. Higher triglyceride and lower HDL-C levels increased the risk of incident CKD. A significant interaction was observed between hypertriglyceridemia and genetic risk factors. The risk of triglyceride-related incident CKD was higher in the low-PRS group. Prospective studies are needed to determine whether lowering triglyceride levels in the general population may reduce the incidence of CKD.

## Supporting information

**S1 Fig. Distribution of each lipid levels represented as quartiles.**
(PDF)

**S2 Fig. Restricted cubic spline curves for each lipid level.** Hazard ratios are estimated using reference values for optimal levels in the 2015 Korean Guidelines for the Management of Dyslipidemia, which were 200 mg/dL mg/dL total cholesterol, 100 mg/dL LDL-C, 60 mg/dL HDL-C, and 150 mg/dL triglycerides.
(PDF)

**S1 Table. Comparison of performance for each PRS algorithm.**
(PDF)

**S2 Table. Univariable analysis using Cox regression model for incident CKD.**
(PDF)

**S3 Table. Subgroup analysis for incident CKD stratified by statin use.**
(PDF)

**S4 Table. Baseline characteristics of study population without history of dyslipidemia or statin use stratified by LDL-C and total cholesterol.**
(PDF)

## Acknowledgments

We thank all the participants who voluntarily enrolled in this study.

## Author Contributions

**Conceptualization:** Jung Pyo Lee, Jeonghwan Lee.

**Data curation:** Boram Weon, Yunjeong Jang.

**Formal analysis:** Boram Weon, Yunjeong Jang, Seounguk Ha.

**Investigation:** Wencheng Jin, Ara Ko.

**Methodology:** Sungho Won, Jeonghwan Lee.

**Project administration:** Jung Pyo Lee, Sungho Won, Jeonghwan Lee.

**Resources:** Sungho Won, Jeonghwan Lee.

**Software:** Jung Pyo Lee, Jeonghwan Lee.

**Supervision:** Yun Kyu Oh, Chun Soo Lim, Jung Pyo Lee.

**Validation:** Jinyeon Jo.

**Visualization:** Boram Weon, Yunjeong Jang.

**Writing – original draft:** Boram Weon.

**Writing – review & editing:** Yunjeong Jang, Sungho Won, Jeonghwan Lee.

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
