## [Decision Letter · Decision Letter 0]

9 Oct 2023

PONE-D-23-23585Association between dyslipidemia and the risk of incident chronic kidney disease affected by genetic susceptibility: polygenic risk score analysisPLOS ONE

Dear Dr. Lee,

Thank you for submitting your manuscript to PLOS ONE. After careful consideration, we feel that it has merit but does not fully meet PLOS ONE’s publication criteria as it currently stands. Therefore, we invite you to submit a revised version of the manuscript that addresses the points raised during the review process.

ACADEMIC EDITOR:

The manuscript is interesting but will require further reworking and a major revision.<o:p></o:p>

While they recognize the potential interest of the subject studied, the reviewers raised a number of important issues that need to be properly addressed.

We look forward to receiving your revised manuscript.

Kind regards,

Marcelo Arruda Nakazone, M.D., Ph.D.

Academic Editor

PLOS ONE

Journal Requirements:

3. We notice that your supplementary [Supplemental Figures 1 and 2] are included in the manuscript file. Please remove them and upload them with the file type 'Supporting Information'. Please ensure that each Supporting Information file has a legend listed in the manuscript after the references list.

Reviewers' comments:

Reviewer's Responses to Questions

**Comments to the Author**

1. Is the manuscript technically sound, and do the data support the conclusions?

Reviewer #1: Yes

Reviewer #2: Partly

2. Has the statistical analysis been performed appropriately and rigorously? 

Reviewer #1: Yes

Reviewer #2: Yes

3. Have the authors made all data underlying the findings in their manuscript fully available?

Reviewer #1: Yes

Reviewer #2: Yes

4. Is the manuscript presented in an intelligible fashion and written in standard English?

Reviewer #1: Yes

Reviewer #2: Yes

5. Review Comments to the Author

Reviewer #1: The manuscript titled "Association between dyslipidemia and the risk of incident chronic kidney disease affected by genetic susceptibility: polygenic risk score analysis" delves into the intricate relationship between dyslipidemia, genetic predisposition (quantified using a polygenic risk score, or PRS), and the onset of chronic kidney disease (CKD). Drawing data from the extensive UK Biobank cohort, the study meticulously constructs a PRS to encapsulate individual genetic susceptibility. Through multivariable Cox regression models, the authors unveiled that heightened triglyceride levels and diminished levels of total cholesterol, LDL-C, and HDL-C amplify the risk of CKD. Moreover, the PRS emerged as a potent predictor of CKD, with a notable interaction between triglyceride levels and PRS, especially in the low-PRS group, signifying a heightened CKD risk.

This research bridges genetic markers and phenotypic indicators to shed light on CKD risk. The methodological rigor, combined with the vastness of the dataset, instills confidence in the findings. However, while the core analysis is robust, the manuscript could benefit from a deeper exploration of underlying mechanisms and broader implications, ensuring the results resonate more profoundly within clinical and research landscapes.

1 - The introduction effectively provides a broad context of the topic at hand. However, the inconsistencies related to dyslipidemia's impact on renal function could be more clearly elucidated. Additionally, delving deeper into the clinical implications, particularly those concerning statin interventions, would enrich the discussion. To enhance the scientific rigor of the introduction, it's essential to emphasize the study's significance, highlight gaps in the existing literature, and clearly outline the foundational rationale behind the research.

2 - Considering the comprehensive methodology you outlined for PRS construction, how did the LDpred2 algorithm's performance compare to the other four algorithms in terms of specificity, sensitivity, and overall accuracy? Could you provide a table or figure that visually contrasts the performance metrics of each algorithm, and elaborate on the primary factors that led to the selection of LDpred2 as the optimal PRS model over the others?

3 - How might the exclusions, particularly those concerning ethnicity and relatedness, affect the broader applicability of the study's findings to diverse populations? Considering the exclusions based on non-white ethnicity and relatedness up to the 3rd degree, have you considered conducting a Principal Component Analysis (PCA) to illustrate the ancestral background of both the included and excluded participants? Such an analysis could provide insights into the genetic diversity of the cohort and further contextualize the potential generalizability of your findings.

4 - Your study unveils the potential of PRS in capturing the risk of incident kidney diseases and emphasizes its interaction with lipid levels, particularly triglycerides. Given this novel insight, are there plans to further investigate the specific genetic loci or pathways that might be mediating this interaction?

5 - Would it be possible to produce a Manhattan plot that illustrates the risk alleles (SNPs) linked to incident CKD within the UK Biobank cohort?

6 - Given that the UK Biobank's population is predominantly white, how imperative is it to replicate this study across diverse ethnic groups to validate the universal applicability of your findings?

Reviewer #2: This is an interesting study that evaluated the association between lipid levels, polygenic risk score and incident CKD in a large population. The authors found that high triglyceride was associated with incident CKD in the low-PRS group.

This association was significant after adjustment to diabetes and statin use, but the model was not adjusted to fibrate use. This is the main limitation of the study knowing that fibrate use for hypertrigyceridemia can induce an increase in serum creatinine. Could this confounding variable be added to the regression model?

6. PLOS authors have the option to publish the peer review history of their article (what does this mean?). If published, this will include your full peer review and any attached files.

Reviewer #1: No

Reviewer #2: No

---

## [Author Response · Author response to Decision Letter 0]

3 Dec 2023

We thank the reviewers for the insightful comments and valuable suggestions. We have revised the manuscript accordingly after the first decision of major revision. Our response to the comments from the reviewers is as follows, and for detailed information, please refer to the attached 'Rebuttal letter.docx.'.

Reviewer #1:

#1-1

Thank you for the important comment. Following the valuable comments from the reviewer, we have added additional descriptions in the Introduction section to emphasize the significance of the present study based on inconsistencies in the existing literature.

#1-2

: Thank you for the valuable comment. Following the comment, we present a comparison of each PRS algorithm for selecting optimal PRS as a supplemental table. We utilized five different algorithms: P+T, C+T, LDpred2, LASSOsum, and PRScs, to calculate the PRS. Specifically, for LDpred2 PRS, we employed a grid model (LDpred2_grid) with proportions of causal variants set at 0.03, 0.01, 0.3, 0.1, and 1, and also utilized an infinitesimal model (LDpred2_inf) and an auto model (LDpred2_auto). Supplemental Table 1 (S1 table) visualizes the performance comparison for each PRS. 

After careful evaluation, we have selected LDpred2_grid_0.03 as the optimal PRS, which demonstrated the highest correlation (R=0.061), significance (p=1.18E-43), and the lowest Akaike information criterion (AIC = 25199.84). In the Method section, we added the description explaining this process. 

#1-3

Thank you for your valuable suggestion. The primary objective of our manuscript is to explore the impact of genetic factors on the relationship between dyslipidemia and chronic kidney disease (CKD) risk, using polygenic risk scores (PRS) in a European population. We utilized the UK Biobank data, which encompasses a large number of subjects, and therefore chose to focus exclusively on the European population to enhance the robustness of our analyses. Additionally, in light of your comment, we reviewed the multidimensional scaling (MDS) plot and concluded that focusing on the European population is indeed the most suitable approach for our study.

#1-4

Thank you for the insightful comment. The reviewer highlighted the necessity of investigating specific risk loci or pathways involved in the interaction between dyslipidemia and genetic risk for CKD. While the reviewer’s suggestion is indeed important, it is considered that an extensive analysis in this direction would require resources beyond the current scope of our research. However, recognizing the significance of this intriguing topic, further investigation is deemed necessary to elucidate and expand upon it. We added the descriptions that explains the rationale behind these needs.

#1-5

Thank you for your valuable comment. The primary objective of our research is to examine the interaction between genetic factors for CKD, manifested by PRS, and the influence of dyslipidemia on CKD. Manhattan plots for CKD risk alleles have already been presented in several GWAS utilizing various cohorts. Consequently, we acknowledge that providing additional Manhattan plots in our study may have a relatively lower impact in terms of relevance to the research topic and novelty. 

We added the descriptions about the existing GWAS in Introduction section and cited two articles presenting Manhattan plots for CKD risk alleles as references.

#1-6

Thank you for your important comment. The reviewer highlighted the necessity for validation regarding the universal applicability across diverse ethnic groups. The authors fully agree with these concerns. Consequently, we are currently undertaking a PRS study utilizing the East Asian population from the Korean Genome and Epidemiology Study (KoGES) and BioBank Japan (BBJ). In addition, we are trying to develop a multi-ethnicity CKD PRS by simultaneously utilizing GWAS summary statistics from Asian and Western populations. Unfortunately, we have not established yet the full datasets, so it is not practical to validate the results of this study in other ethnic groups in the limited timeframe available. We've added these limitations to the Discussion section.

Reviewer #2:

#2-1

Thank you for the valuable comment. The authors acknowledge these concerns. We conducted a re-analysis by incorporating a variable of fibrate use into the multivariable model, and additionally included fasting time as additional covariate, considering the significant relationship between fasting status and triglyceride level. Subsequent re-analysis results revealed that the impact of each lipid level on CKD and the interactions between lipid levels and PRS were largely consistent with the initial analysis. Consequently, corresponding adjustments were made to Table 1, 2, 3, 4, and Figure 2 to reflect the updated content. The description of these findings in the Results section has also been revised.

---

## [Decision Letter · Decision Letter 1]

16 Jan 2024

PONE-D-23-23585R1Association between dyslipidemia and the risk of incident chronic kidney disease affected by genetic susceptibility: polygenic risk score analysisPLOS ONE

Dear Dr. Lee,

Thank you for submitting your manuscript to PLOS ONE. After careful consideration, we feel that it has merit but does not fully meet PLOS ONE’s publication criteria as it currently stands. Therefore, we invite you to submit a revised version of the manuscript that addresses the points raised during the review process.

**ACADEMIC EDITOR: **

Previous comments have been addressed, but the manuscript will require minor revisions. While recognizing the potential interest of the subject studied, one of the reviewers raised additional comments that need to be properly addressed.

We look forward to receiving your revised manuscript.

Kind regards,

Marcelo Arruda Nakazone, M.D., Ph.D.

Academic Editor

PLOS ONE

Journal Requirements:

Reviewers' comments:

Reviewer's Responses to Questions

**Comments to the Author**

1. If the authors have adequately addressed your comments raised in a previous round of review and you feel that this manuscript is now acceptable for publication, you may indicate that here to bypass the “Comments to the Author” section, enter your conflict of interest statement in the “Confidential to Editor” section, and submit your "Accept" recommendation.

Reviewer #2: All comments have been addressed

Reviewer #3: All comments have been addressed

2. Is the manuscript technically sound, and do the data support the conclusions?

Reviewer #2: Yes

Reviewer #3: Partly

3. Has the statistical analysis been performed appropriately and rigorously? 

Reviewer #2: Yes

Reviewer #3: Yes

4. Have the authors made all data underlying the findings in their manuscript fully available?

Reviewer #2: Yes

Reviewer #3: Yes

5. Is the manuscript presented in an intelligible fashion and written in standard English?

Reviewer #2: Yes

Reviewer #3: Yes

6. Review Comments to the Author

Reviewer #2: Thank you.

The authors addressed all my concerns.

I have no further comments.

Reviewer #3: Thanks for your work on an interesting topic. You should underline more the limitations of the UK Biobank as history of diabetes and history of diabetes aren't sufficient enough to reflect the level of BP and/or glycemic time-exposure which should play a role in the occurence of kidney disease. Therefore the discussion should provide a complete overview of all biases for the readers, underlying that further specific research is warranted.

7. PLOS authors have the option to publish the peer review history of their article (what does this mean?). If published, this will include your full peer review and any attached files.

Reviewer #2: No

Reviewer #3: **Yes: **Pierre SABOURET

---

## [Author Response · Author response to Decision Letter 1]

6 Feb 2024

<Response to Editor> 

Thank you for the comment. We have reviewed the reference list and found a correction in reference number 31. This correction was due to an error in electronic tagging and did not change the content of the article. However, after review, we changed the reference to a more appropriate reference and made the following changes to the reference list of the manuscript: 

- Original Reference 31: Ebrahim S, Sung J, Song YM, Ferrer RL, Lawlor DA, Davey Smith G. Serum cholesterol, haemorrhagic stroke, ischaemic stroke, and myocardial infarction: Korean national health system prospective cohort study. Bmj. 2006;333(7557):22. Epub 2006/06/08. http://doi.org/10.1136/bmj.38855.610324.80

- Revised Reference 31: Miller M. Dyslipidemia and cardiovascular risk: the importance of early prevention. QJM. 2009;102(9):657-67. Epub 2009/06/04. http://doi.org/10.1093/qjmed/hcp065

<Response to Reviewer #3> 

Thank you for the valuable comment. In response the comment, we further explained the limitations of not having a time exposure factor for diabetes and hypertension in the Discussion section, emphasizing the need for further research: “Furthermore, information regarding the timing of diagnosis of diabetes and hypertension was not available from the UK Biobank, variables regarding the duration of exposure to hyperglycemia and high blood pressure, which are important in the development of kidney disease, could not be applied to the analysis. Further cohort studies, including more precise and regular assessments of clinical and lifestyle data incorporating temporal information, are warranted.”

---

## [Decision Letter · Decision Letter 2]

14 Feb 2024

Association between dyslipidemia and the risk of incident chronic kidney disease affected by genetic susceptibility: polygenic risk score analysis

PONE-D-23-23585R2

Dear Dr. Lee,

We’re pleased to inform you that your manuscript has been judged scientifically suitable for publication and will be formally accepted for publication once it meets all outstanding technical requirements.

Kind regards,

Marcelo Arruda Nakazone, M.D., Ph.D.

Academic Editor

PLOS ONE

Additional Editor Comments (optional):

Reviewers' comments:

Reviewer's Responses to Questions

**Comments to the Author**

1. If the authors have adequately addressed your comments raised in a previous round of review and you feel that this manuscript is now acceptable for publication, you may indicate that here to bypass the “Comments to the Author” section, enter your conflict of interest statement in the “Confidential to Editor” section, and submit your "Accept" recommendation.

Reviewer #3: All comments have been addressed

2. Is the manuscript technically sound, and do the data support the conclusions?

Reviewer #3: Yes

3. Has the statistical analysis been performed appropriately and rigorously? 

Reviewer #3: Yes

4. Have the authors made all data underlying the findings in their manuscript fully available?

Reviewer #3: Yes

5. Is the manuscript presented in an intelligible fashion and written in standard English?

Reviewer #3: Yes

6. Review Comments to the Author

Reviewer #3: Dear author, you clearly answered to all comments.

This article may be accepted for a publication in PlosOne.

Further research is warranted to better define the causation btw lipid profiles, PRS and CKD.

Congratulations for your work.

7. PLOS authors have the option to publish the peer review history of their article (what does this mean?). If published, this will include your full peer review and any attached files.

Reviewer #3: **Yes: **SABOURET Pierre

---

## [Editor Report · Acceptance letter]

2 Apr 2024

PONE-D-23-23585R2 

PLOS ONE

Dear Dr. Lee, 

I'm pleased to inform you that your manuscript has been deemed suitable for publication in PLOS ONE. Congratulations! Your manuscript is now being handed over to our production team.

Kind regards, 

on behalf of

Professor Marcelo Arruda Nakazone 

Academic Editor

PLOS ONE